# Topical Application of Linezolid–Loaded Chitosan Nanoparticles for the Treatment of Eye Infections

**DOI:** 10.3390/nano13040681

**Published:** 2023-02-09

**Authors:** Musaed Alkholief, Mohd Abul Kalam, Abdullah K. Alshememry, Raisuddin Ali, Sulaiman S. Alhudaithi, Nasser B. Alsaleh, Mohammad Raish, Aws Alshamsan

**Affiliations:** 1Department of Pharmaceutics, College of Pharmacy, King Saud University, P.O. Box 2457, Riyadh 11451, Saudi Arabia; 2Department of Pharmacology and Toxicology, College of Pharmacy, King Saud University, P.O. Box 2457, Riyadh 11451, Saudi Arabia

**Keywords:** linezolid, chitosan, nanoparticles, antibacterial, eye–irritation, transcorneal–permeation, Gram–positive, Ocular–bioavailability

## Abstract

Linezolid (LZ) loaded chitosan–nanoparticles (CSNPs) was developed by the ionic–gelation method using Tripolyphosphate–sodium as a crosslinker for topical application for the treatment of bacterial eye infections. Particles were characterized by Zeta–Sizer (Malvern Nano–series). TEM was used for structural morphology. Encapsulation and drug loading were estimated by measuring the unencapsulated drug. In-vitro drug release in STF (pH 7) was performed through a dialysis membrane. Storage stability of LZ–CSNPs was checked at 25 °C and 40 °C for six months. The antimicrobial potency of NPs was evaluated on different Gram–positive strains. Ocular irritation and pharmacokinetic studies were completed in rabbits. Ex-vivo transcorneal permeation of the drug was determined through the rabbit cornea. Ionic interaction among the oppositely charged functional groups of CS and TPP generated the CSNPs. The weight ratio at 3:1, wt/wt (CS/TPP) with 21.7 mg of LZ produced optimal NPs (213.7 nm with 0.387 of PDI and +23.1 mV of ZP) with 71% and 11.2% encapsulation and drug loading, respectively. Around 76.7% of LZ was released from LZ–AqS within 1 h, while 79.8% of LZ was released from CSNPs at 12 h and 90% at 24 h. The sustained drug release property of CSNPS was evaluated by applying kinetic models. The linearity in the release profile suggested that the release of LZ from CSNPs followed the Higuchi–Matrix model. LZ–CSNPs have shown 1.4 to 1.6-times improved antibacterial activity against the used bacterial strains. The LZ–CSNPs were “minimally–irritating” to rabbit eyes and exhibited 4.4-times increased transcorneal permeation of LZ than from LZ–AqS. Around 3-, 1.2- and 3.1-times improved T_max_, C_max_, and AUC_0–24 h_, respectively were found for LZ–CSNPs during the ocular pharmacokinetic study. AqS has shown 3.1-times faster clearance of LZ. Conclusively, LZ–CSNPs could offer a better alternative for the prolonged delivery of LZ for the treatment of bacterial infections in the eyes.

## 1. Introduction

Eye infections due to different strains of Gram–positive pathogens including the methicillin–resistant *Staphylococcus aureus* (MRSA) strains are vision–threatening and a potential clinical challenge to physicians [1,2]. Linezolid (LZ) is the first licensed drug that belongs to the oxazolidinone class of antibiotics and can be a choice of therapeutic agent to treat such infections and also in the eyes undergoing elective cataract surgery [3,4]. It is a broad–spectrum antibiotic, primarily active against Gram–positive pathogens, including both methicillin–sensitive and methicillin–resistant *Staphylococcus*, *Enterococci*, *Streptococcus pneumoniae*, etc. The MIC of LZ against such microorganisms ranges from 1–4 µg/mL [5]. In human, LZ shows 100% bioavailability after repeated oral dosing of 650 mg or intravenous administration and give peak serum concentration (C_max_) of around 18 µg/mL, while in animal studies it reaches in the eyes at a concentration of around 7.2 µg/mL (∼40% of serum concentration) after crossing the blood–ocular barrier [4]. These findings suggested that LZ may achieve its therapeutic level in aqueous and vitreous chambers after repeated oral/intravenous dosing as an antibiotic therapy against serious bacterial infections including the Gram–positive endophthalmitis [3]. Although vancomycin can be the choice of drug to treat such infections, its effectiveness has been compromised because of the emergence of resistant strains [6].

The repeated oral/intravenous administration of LZ to treat eye infections may give rise to drug resistance, adverse drug reactions, and side effects. Additionally, the topical ocular application limits the ocular bioavailability of drugs because of naturally strong defensive eye barriers. Noncorneal absorption, nasolacrimal drainage, and corneal impenetrability [7] cannot allow more than 7% of the applied dose to be available for eye tissues [8,9,10]. The ocular drug availability can be increased by extending the precorneal retention of the applied dosage forms and augmenting the conjunctival and corneal drug transport. In some eye conditions, frequent dosing is required which can cause some unwanted phenomena such as sensitivity, corneal pigmentation, and some mechanical injuries [11].

Therefore, it was requisite to develop a topically applied novel formulation of LZ, which would limit the frequent dosing to avoid the unwanted effects of LZ. Encapsulation and loading of drugs into nanocarriers can be a good way to avoid some shortcomings of conventional eye drops [11,12,13]. Thus, we develop non–invasive nanocarriers (CSNPs) for topical ocular delivery of LZ to treat the increasing multi–drug–resistant eye infections. The CSNPs prolong the ocular surface retention, enhance the transcorneal–flux, and ultimately the intraocular availability of the drug [14,15,16].

Chitosan (CS) was used as a polymer in the preparation of CSNPs, because of its non–toxic and biodegradable properties. It is a hydrophilic and mucoadhesive polysaccharide [17,18]. Chitosan stabilizes the tear fluids, enhances ocular retention, reduces the nasolacrimal drainage (due to sufficient viscosity and high adhesion) of CSNPs, and consequently, improves the ocular availability of the loaded drug [19,20,21,22]. Chitosan possesses antibacterial activity against some Gram–positive and Gram–negative bacteria [23] and also has some antifungal potency [24,25,26].

The CS has shown sustained intraocular permeation of encapsulated drugs by reversible binding with the corneal epithelium (due to electrostatic interaction with negative mucin layer on the eye surfaces) and loosening its tight junctions [27,28]. The increased drug availability in the eyes and the antibacterial potency of CS together augment the overall antibacterial activity against Gram–positive and MRSA eye infections with less frequent dosing and good patient compliance [16,22,24,27]. Therefore, CS was considered one of the best polymers of biological origin applicable for eye preparations [29]. The TPP was used as a cross–linker for CS because it has the potential for ionic crosslinking of CS at lower pH and by the mechanism of deprotonation at higher pH. At lower pH the complexation of CS–TPP occurs by the ionic interaction between the positively charged amino group (–NH^3+^) of CS and the negatively charged phosphoric ions (–P_3_O_10_^5−^) of TPP [30].

Previous research showed that LZ has efficient antibacterial potency to treat keratitis due to Gram–positive pathogens. The ocular applications of LZ–loaded CSNPs are still not explored well. Therefore, here we developed and characterized LZ–CSNPs for their ocular suitability. The antibacterial activity (in vitro) of LZ from CSNPs was compared to the LZ–AqS against some Gram–positive pathogens including an MRSA strain. Transcorneal permeation of the drug from the LZ–containing formulations was evaluated across the excised rabbit cornea and eye irritation of the formulations was tested in rabbits. Finally, the ocular pharmacokinetics of LZ from CSNPs was performed as compared to LZ–AqS in rabbits.

## 2. Materials and Methods

### 2.1. Materials

Linezolid (C_16_H_20_FN_3_O_4_; MW. 337.3 g/mol) of a ≥99% purity was purchased from “Beijing Mesochem Technology Co. Ltd. (Beijing, China)”. “Sodium–Tripolyphosphate (TPP), low MW chitosan (50–190 kDa, 75–85% deacetylated), and sodium acetate (CH_3_COONa) were purchased from Sigma Aldrich (St. Louis, MO, USA)”. “HPLC grade methanol, acetonitrile, and acetic acid glacial were procured from BDH, Ltd. (Poole, England)”. “The dialysis membrane of 12–14 kDa (MWCO) was purchased from Spectra/Por^®^, Spectrum Laboratories Inc., (Rancho Dominguez, CA, USA)”. “Water (purified) was obtained from Milli–Q^®^ water purification system (Millipore, France). “All other used chemicals and solvents were of analytical and HPLC grades, respectively.

### 2.2. Chromatographic Analysis of LZ

The HPLC–UV system (at 254 nm UV–detection) was used for the analysis of LZ by adopting the reported methods [31,32,33]. Briefly, the Waters^®^ HPLC system (1500-series controller, Milford, MA, USA) equipped with Waters^®^ UV–detector (Model-2489, dual absorbance detector, USA), a Waters^®^ binary pump (Model-1525, USA), Waters^®^ automated sampling system (Model-2707, USA) was used. The “Breeze software” was used to monitor the system. A reverse phase C_18_ analytical column (250 mm × 4.6 mm, 5 μm particle size) manufactured by “Macherey–Nagel” at 40 °C was used for the LZ analysis. The mobile phase consisted of 65: 35 (*v*/*v*) of 0.02 M sodium acetate (CH_3_COONa) buffer (pH was maintained to 3.5 using O–phosphoric acid) and acetonitrile was isocratically pumped at the flow rate of 1 mL/min. The injection volume was 20 µL and the total run time was 8 min. A standard stock solution of LZ was prepared in acetonitrile (100 μg/mL), while working standard dilutions of 0.25–50 μg/mL concentration ranges were obtained by serial dilution of the standard stock solution with the mobile phase mixture.

### 2.3. Formulation Development

The ionic–gelation method was used to prepare the LZ–loaded CSNPs, where TPP was used as a cross–linker [34]. Briefly, 21.7 mg of LZ was dissolved in 27 mL of CS solution (0.6 mg/mL), where CS solution was prepared in 1% *v*/*v* of glacial acetic acid. Then, 13 mL of TPP solution (0.4 mg/mL) in Milli–Q^®^ (Millipore, Molsheim, France) water was added drop by drop (at 2 mL/min) into the LZ containing CS solution while maintaining the mixture at 700 rpm of constant magnetic stirring (for 3–4 h) at ambient temperature. The excess amount of drug (that might not have been encapsulated) was washed–out by centrifugation at 13,000 rpm and 10 °C for 20 min. Thereafter, the drug–loaded CSNPs were collected by ultracentrifugation with Milli–Q^®^ water (three times) at 25,000 rpm and 4 °C for 35 min. The obtained pellets of LZ–CSNPs were suspended in water, filtered through a 450 µ filtration unit, frozen at −70 °C, and freeze–dried (for 24 h at −50 °C and 0.02 mbar). The cryoprotectant (mannitol at 1%, *w*/*v*) was mixed into the CSNPs–suspension before freeze–drying [35]. Finally, the freeze–dried products were stored at −20 °C for further experiments.

### 2.4. Characterization of LZ–Loaded CSNPs

#### 2.4.1. Particle Characterization

The hydrodynamic diameter, polydispersity–index (PDI), and surface charge (zeta–potential) of LZ–CSNPs were determined by the Dynamic Light Scattering technique. It was performed at a detection arrangement of 90° at 25 °C by Zetasizer Nano–Series (Nano–ZS, Malvern Ins. Ltd., Worcestershire, United Kingdom). The nanosuspension was further diluted with Milli–Q^®^ water (having a dielectric constant of ≈78.5) as a dispersant for the above measurements. The above processes were performed by a software DTS V–4.1 (Malvern, UK) equipped with the system, and the measurements were performed in triplicate.

#### 2.4.2. Transmission Electron Microscopy (TEM)

The structural morphology of LZ–CSNPs was checked by “TEM, JEM–1010 (JEOL, Tokyo, Japan)”. The analysis was completed under light microscopy, which had the capability of point-to-point resolution and operated at 80 kV. The magnification of the image was 100 K–times. The combination of bright field microscopy with increasing magnifications was applied for the interpretation of the CSNPs’ structure. The nanosuspension of the NPs was additionally diluted with purified water before the analysis. A drop of nanosuspension was placed on the copper grids (which were coated with carbon), and a drop of 2% solution of phosphotungstic acid was put on the drop of nanosuspension for staining purposes. The copper grids were left overnight for air drying. Thereafter, the particle morphology was observed under transmission electron microscopy at room temperature.

### 2.5. Drug Encapsulation and Loading Efficiencies

The encapsulation and loading of the drug in the CSNPs were calculated by quantification of the unencapsulated LZ (indirect method). The percentage of encapsulation (*EE*%) and loading (*DL*%) were calculated by the differences between the initial quantity of LZ to prepare the LZ–CSNPs and the quantity of LZ quantified in the supernatant. Briefly, 10 mg of CSNPs was dispersed in methanol, vortexed for 30 s, and centrifuged at 13,000 rpm at a low temperature for 10 min. The supernatant was collected and around 20 µL was injected into the HPLC–UV system to analyze LZ [31,32,33]. The *EE*% and *DL*% were calculated by the following expressions Equations (1) and (2):(1)$$EE\%=\left(\frac{Initial\;quantity\;of\;drug\;used\;\left(µg\right)-Quantity\;of\;drug\;in\;the\;supernatant\;(µg)}{Initial\;quantity\;of\;drug\;used\;\left(µg\right)\;}\right)\times 100$$
(2)$$DL\%=\left(\frac{Initial\;quantity\;of\;drug\;used\;\left(µg\right)-Quantity\;of\;drug\;in\;the\;supernatant\;(µg)}{Total\;amount\;of\;drug\;loaded\;CSNPs\;\left(µg\right)}\;\right)\times 100$$

### 2.6. Physicochemical Characterization

These characterizations would ensure the suitability of the CSNPs for ophthalmic use. The clarity/transparency of nanosuspension of LZ–CSNPs by visual observation under light alternatively against white and black backgrounds after adjusting the pH of the nanosuspension to 7.2. Other parameters including the pH and osmolarity of the nanosuspension were checked by pH meter (Mettler Toledo MP–220, Greifensee, Switzerland) and Osmometer (Fiske Associates, Waterford, PA, USA), respectively. The viscosity was determined by Sine–wave Vibro Viscometer (Model SV–10, A & D Co. Ltd. Tokyo, Japan) at ≈35 ± 0.5 °C (ocular physiological) and ≈25 ± 0.5 °C (non-physiological) temperatures [36,37].

### 2.7. In Vitro Drug Release and Kinetics of Drug Release

The dialysis bag (MWCO: 12–14 kDa) was used as the release barrier. The isotonic nanosuspension of LZ–CSNPs and LZ–AqS (control) were subjected to in vitro release study using STF as a release medium. The 6.8 g of NaCl, 2.2 g of NaHCO_3_, 1.4 g of KCl, and 0.08 g of CaCl_2_.2H_2_O were dissolved in 1000 mL purified water to prepare the STF (pH 7.4). An equivalent volume of nanosuspension of LZ–CSNPs and LZ–AqS (containing 1000 µg of LZ) was filled into the pre–activated dialysis bags and the ends of the bags were closed using closures. The dialysis bags were put into beakers containing STF (50 mL each). The set-ups were placed in a shaking (100 rpm) water bath maintained at 35 ± 1 °C. At fixed time intervals, 1 mL of each sample was collected from each beaker, and after sampling equal volume of fresh STF was added to each beaker. The collected samples were centrifuged and 20 µL of the supernatant was injected into the HPLC–UV system to determine the drug concentration. The LZ–AqS was formulated by suspending LZ (10 mg) in 10 mL of 0.25% (*w*/*v*) Polysorbate–20 aqueous solution [38,39]. The experiment was executed three times for each LZ–formulations. The cumulative amount of drug released (DR%) was calculated by the following expression Equation (3).
(3)$$DR\%=\frac{Conc.\left(µg/\mathrm{mL}\right)\times DF\times Volume\;of\;release\;medium\;\left(\mathrm{mL}\right)}{Initial\;quantity\;of\;LZ\;used\;for\;the\;experiment\;(µg)}\times 100$$
where “*DF*” is the dilution factor. The release data were fitted to different kinetic models “(Zero–order, First–order, Higuchi–Matrix Square–Root, Hixson–Crowell Cube–Root, and Korsmeyer–Peppas)”. The best–fitted kinetic model for LZ release from CSNPs was categorized based on the highest value of the coefficient of correlation (*R*^2^). From the slopes and intercepts of the different release plots, the release exponent (*n*-value) was calculated [40]. The *n*-value would suggest the mechanism of LZ release from CSNPs [13,41,42].

### 2.8. Antimicrobial Study

Antimicrobial activity and minimum inhibitory concentrations (MICs) of LZ–CSNPs and its counter formulation (LZ–AqS) were completed by agar diffusion technique [43]. For this, “the bacterial strains were obtained from the Microbiology Unit at the Department of Pharmaceutics, College of Pharmacy, King Saud University”. The strains were chosen from the Global Priority Pathogens List. A total of four ATCC standard strains of *Bacillus subtilis*, *Staphylococcus aureus*, MRSA (SA–6538), and *Streptococcus pneumoniae* were employed for the susceptibility against LZ. Each strain was spread on the separate Mueller Hinton agar plates. After 30 min, three 6 mm diameter wells were made using a sterile borer. In the first well, 50 µL of LZ–AqS (50 µg of LZ), in the second well, an equivalent volume of LZ–CSNPs suspension (≈50 µg of LZ), and in the third well, 50 µL of CSNPs (without LZ) were added. After 1 h, all the inoculated plates were incubated for 24 h at 37 °C, and the zone of inhibitions was measured for each product. The entire evaluation was completed in triplicates.

### 2.9. Stability Study

This study was performed to evaluate the storage stability of LZ–CSNPs in terms of particle size, polydispersity index (PDI), zeta potential (ZP), encapsulation efficiency (EE%), and drug loading (DL%) [13,44]. Accurately weighed (10 mg) freeze–dried samples of LZ–CSNPs were packed in six separate glass containers. Three containers were stored at 25 ± 1 °C and three at 40 ± 1 °C for six months. The changes in the mentioned parameters were checked at one week, one, three, and six months. The stored samples were re-dispersed in STF before each evaluation.

### 2.10. In Vivo Animal Study

The albino rabbits (New Zealand) with a 2.0–3.0 kg body weight were obtained from the “Animal care and use center, College of Pharmacy, King Saud University for the in vivo ocular experiments”. “The ethical approval for the experiments on rabbits was obtained from the King Saud University Research Ethics Committee (amended approval number KSU-SE-18-25)”. All the rabbits were checked for any clinical defects in the eyes, housed as per the recommendation of “Guide for the Care and Use of Laboratory Animals”, and given a standard pellet diet and water *ad-libitum*”. All the animals were kept on fasting overnight before the experiments.

#### 2.10.1. Eye Irritation Study

The eye irritation test was performed as per the “Association for Research in Vision and Ophthalmology (ARVO)” for animal use in “Ophthalmic and Vision Research” guidelines to assess the ocular tolerability of drug–loaded CSNPs. For the method to perform this test we followed Draize’s test protocol in healthy rabbits [45]. As per the ARVO, only one eye (right) of the rabbits was selected for testing the product while the left eye was treated with normal saline (negative control). Six rabbits were divided into two groups each containing three for each test product (LZ–CSNPs and blank–CSNPs) [46]. For acute irritation, 40 μL of formulations (three repeated doses at every 15 min) was put into the conjunctival sac of the right eyes of all the animals of respective groups, and normal saline was instilled into the left eyes. Around 1 h of exposure, the eyes were examined visually for any unwanted signs and symptoms in the cornea, iris and conjunctiva or any other changes in the formulation treated eyes as compared to the saline treated ones. The photographs of the eyes were captured at different time–intervals for scoring and the irritation level was assessed as per the scoring guideline based on the discomfort, signs, and symptoms including redness, swelling, chemosis in the eye structures, or any mucoidal discharge [28,47]. The scoring was completed and irritation due to the test products (if any) was classified [48,49].

#### 2.10.2. Transcorneal Permeation

After a washout period of three weeks, the rabbits used for the irritation study were sacrificed with an overdose injection of Ketamine. HCl and Xylazine mixture. The eyes were enucleated, and the cornea was excised. Transcorneal permeation of LZ from CSNPs as compared to LZ–AqS was performed using “double–jacketed diffusion cells equipped with automated sampling system–SFDC 6, LOGAN, New Jersy, USA”. The excised cornea was fixed between the receptor and donor counterparts of diffusion cells in a way where the corneal epithelial layer encountered the donor part of the cell. The receptor part was filled with STF; a small magnetic bar was put into it to remove unwanted air bubbles during the experiment. The diffusion cells were placed on the LOGAN instrument and water at 37 ± 1 °C was flown in the outer jacket of the cells. For each product (in triplicate), 500 μL of LZ–AqS (500 µg of LZ) and an equivalent amount of LZ–CSNPs (~500 µg of LZ) were transferred into the donor parts, then the instrument was started. The sampling was completed from the receptor parts at different time points and the concentration of drug permeated through the cornea was analyzed by the HPLC–UV method [31,33].

The amount of drug crossed the cornea was deliberated in view of the volume of STF in the receptor part (5.2 mL), the cross–section area of diffusion cell or involved cornea (0.502 cm^2^), and the initial drug concentration (*C*_0_). The permeated amount (μg/cm^2^) through the cornea was calculated by expression Equation (4) and plotted against the time.
(4)$$Amount\;of\;drug\;permeated\;\left(µg/{\mathrm{cm}}^{2}\right)=\frac{Conc.\left(µg/\mathrm{mL}\right)\times DF\times Volume\;of\;receptor\;part\;\left(\mathrm{mL}\right)}{Crosse\;section\;area\;of\;diffusion\;cell\;({\mathrm{cm}}^{2})}$$
where “*DF*” was the dilution factor. The slope of the plot was utilized to calculate the permeation parameters by the following expressions Equations (5) and (6).
(5)$$J\;\left(µg/{\mathrm{cm}}^{2}/h\right)=\frac{dA}{dt}$$
(6)$$P_{app}\;\left(\mathrm{cm}/h\right)=\frac{J}{C_{0}}$$
where “*J*” is the steady–state flux, “*P_app_*” is the apparent permeability, “*A*” is the amount of LZ crossed through the cornea, “*^dA^/_dt_*” is the linear ascent of the slope, “*t”* is the corneal contact time of formulations and “*C*_0_” is the initial concentration of drug in the donor portion of the cells.

#### 2.10.3. Ocular Bioavailability

The ocular availability of LZ was determined, after the topical application of LZ–containing formulations in the eyes of healthy rabbits. Six animals were divided into two groups (the first group for LZ–CSNPs and the second group for LZ–AqS). Around 40 μL of sterilized LZ–containing formulations (equivalent to 40 µg of LZ) were instilled in the right eyes of the respective group rabbits. After 1 h of dosing, the animals were sedated with intravenous administration of the mixture of Ketamine. HCl (15 mg/kg of b. wt.) and Xylazine (3 mg/kg of b. wt.) [3,16,35,47,50]. Successively,50 µL of AqH was taken out by a 29-gauge needle attached to a tuberculin syringe at predetermined time intervals. The obtained AqS samples were kept at −70 °C till the analysis was completed. The samples for the analysis were prepared and the drug concentrations in the samples were analyzed by the HPLC–UV method [31,33].

#### 2.10.4. Chromatographic Separation of LZ in Aqueous Humor (AqH)

The chromatographic separation of LZ was completed by adopting a reported method [31]. The chromatographic conditions and instrumentations have been briefed as above. For the analysis of LZ in biological fluid samples (AqH samples), tedizolid (TDZ) was used as an internal standard (IS). The standard stock solution of LZ was prepared in acetonitrile (1000 ng/mL), while working standard dilutions of 5–1000 ng/mL concentration ranges were obtained by serial dilution of the standard stock solution with the mobile phase mixture. Similarly, an accurate amount of TDZ was dissolved in DMSO (200 µL) and diluted with methanol to get a stock solution of IS (1000 ng/mL). The stock solution of IS was further diluted with methanol to obtain (100 ng/mL of working solution).

The AqH samples stored at −70 °C were thawed at ambient temperature and into each 50 μL sample, 25 μL of the IS working solution was spiked and vortexed in 500 μL capacity Eppendorf tubes. Then, 325 μL acetonitrile (for protein precipitation) was added into each Eppendorf tube and vortexed again for 30 s. The vortexed and mixed samples (400 μL) were centrifuged at 13,000 rpm for 15 min, supernatants were collected, and transferred UPLC max recovery vials. Finally, 20 μL of the supernatant was injected into the HPLC–column for the chromatographic analysis of the drugs.

### 2.11. Data Analysis

All the data were represented as the mean of three measurements with standard deviation (Mean ± SD) unless otherwise indicated. The pharmacokinetic parameters were calculated by a non–compartmental approach using “*PK*–Solver software, Nanjing, China in MS-Excel-2013” [51]. The statistical analysis was completed and graphs were plotted using “GraphPad Prism: V–5 (GraphPad Software, Inc., San Diego, CA, USA)”. The results were compared by Student’s *t*-test by taking “*p* < 0.05” as statistically significant.

## 3. Results and Discussion

### 3.1. Formulation Development

The LZ–loaded CSNPs were formulated by the ionic–gelation method using TPP as a cross–linker [34]. The LZ–CSNPs were optimized as per our previous reports [16,52]. Here, the optimal concentrations of TPP and CS (0.4 and 0.6 mg/mL, respectively) at 3 h of continuous magnetic stirring (700 rpm) at room temperature with 21.7 mg of LZ produced LZ–loaded CSNPs with desired characteristics (including the minimum size with maximum ZP, EE%, and DL%). The interaction (ionic) between the negatively charged functional groups of TPP and positively charged quaternary amine groups of CS produced optimal NPs at 1:3 weight–by–weight ratio of TPP/CS. Among the different formulations (Appendix A) the best one based on optimal size, high ZP with maximum EE% and DL% was selected for further studies. In brief, at TPP/CS weight ratio of (78 mg/162 mg with 21.7 mg of LZ) and at 700 rpm of stirring rate for 3 h was good to obtain LZ–CSNPs with particle size 209.7 nm (with PDI of 0.387 and ZP of + 23.1 mV) having higher EE% and DL% (71% and 11.2%, respectively). During the formulation development, LZ was easily solubilized in CS solution because of its slightly high solubility in the aqueous phase (3 mg/mL) [33]. With the better aqueous solubility of LZ and the optimal TPP/CS weight ratio, the anionic functional groups present in the structure of TPP properly interacted with the cationic (amine) functional groups of CS and resulted in LZ–CSNPs with improved characteristics.

Due to the hydrophilicity, biodegradability, mucoadhesive nature, and non–toxic and non–irritating (for eyes) properties of CS, it was a choice of natural polymer for many drug delivery applications. Thus, CS was used to prepare the LZ–CSNPs in the present investigation. Due to its non–toxic, non–irritant quality to eyes and better safety profiles, it was anticipated that the use of CS would stabilize the tear fluids in the eyes. Therefore, reduce the unwanted drainage of the LZ–CSNPs and prolong its ocular retention, which in turn would increase the transcorneal permeation, and intraocular penetration of the nano–carrier by interacting with the corneal epithelium, triggering a reversible loosening of corneal epithelial tight junctions and improve the ocular bioavailability of linezolid [8,13,16,53].

### 3.2. Particle Characterization

The characterization of NPs comprises the examination of the structures at the nano–range. The size and shape of NPs or surface adsorbents (if any) are crucial steps to apprehending the relationships among the quality, performance, and safety/toxicity of NPs. The Dynamic Light Scattering technique using Zetasizer was good for the characterization of the particle (size, PDI and ZP). The LZ–loaded CSNPs at TPP/CS weight ratio of 78 mg: 162 mg with 21.7 mg of the drug was showing optimum–sized particles (209.7 ± 8.2 nm) with low PDI value (0.387 ± 0.134) and + 23.1 ± 2.9 mV of zeta–potentials having 71.0 ± 5.2% and 11.2 ± 1.8% of EE% and DL%, respectively. The resultant particle size of the LZ–CSNPs in the present study was suitable for its ocular use as per the size of particulate materials (≤10 μ) that can be easily tolerated by human eyes without any irritation, discomfort, corneal injury/abrasion, or grating of ocular surfaces [54]. The obtained CSNPs with 209.7 nm size would improve patient compliance without causing any discomfort to the eyes during its application. Similarly, the linezolid aqueous suspension (LZ–AqS) prepared by suspending 10 mg of LZ in 10 mL of 0.25% (*w*/*v*) Polysorbate–20 aqueous solution was subjected to DLS measurement. The size of the suspended particles was 800.5 ± 63.5 nm, polydispersity was 0.634 ± 0.027 and zeta potential was −13.5 ± 2.4 mV.

The positive ZP (+ 23.1 mV) of the LZ–CSNPs envisaged the physical stability of the colloidal dispersion of the NPs in its nano–suspension form because the positive surface charges on the NPs would repel each other electrostatically and avoid self–aggregation and settling of the particles. The PDI measures the width of size distribution, a low PDI value (0.387) suggested the unimodal distribution of the LZ–CSNPs in their dispersed state. The size and ZP distribution curves of LZ–CSNPs were presented respectively in Figure 1a,b.

The amounts of TPP (78 mg) and CS (162 mg) with 21.7 mg of LZ has given sufficient production yield with around 71% and 11.2% of drug encapsulation and loading efficiencies, respectively. These findings might be associated with the higher viscosity of the CS solution, which allowed rapid solidification due to the faster crosslinking phenomenon by TPP, which in turn decreased the drug leaching into the dispersion medium. As a result, the increased yield value, drug encapsulation, and loading efficiencies were in agreement with our previous reports [13,16,52].

To validate the suitability of the formulation for ocular use, the LZ–CSNPs were further subjected to TEM for morphological characterization. It is also important if any changes (due to oxidation/reduction) have occurred during the sample preparation for morphological examination of the NPs through transmission electron microscopy [16,55,56]. The nanosuspension of CSNPs was further diluted with purified water before the TEM analysis to avoid certain challenges (image overlapping, difficult detection of very small particles, and concealing signals during the examination due to the presence of background noise and surrounding matrix). The TEM was completed under light microscopy, operated at a relatively low accelerating voltage (80 kV) for high–resolution imaging without any structural damage to the NPs due to the high–energy electron beam. The applied accelerating voltage in the present case was considered low as compared to high–energy electron beams (200–1000 kV) used for metallic particles and intermediate–voltage (200–400 kV) used for biological/non–metallic samples). The TEM image of the LZ–CSNPs has shown that the NPs were spherical in shape, separated from each other without agglomeration with a solid dense structure. Although, mannitol was added to the nanosuspension before freeze–drying the CSNPs. Still, in certain portions of the TEM image (Figure 2), the NPs were aggregated with each other. This unwanted aggregation might be due to the high mucoadhesive nature of chitosan (CS).

### 3.3. Physicochemical Characterization

The clarity/transparency, pH, osmolarity, and viscosity of LZ–containing formulations (LZ–CSNPs and LZ–AqS) were determined, and results were compared with the STF and summarized in Table 1. The formulation was transparent, hence there would not be a blurring of vision. The pH of the formulation was almost equal to the pH of tear fluids, so eye fluids can easily buffer the pH of the formulation to its own pH condition and would prevent any irritation to the eyes related to the pH of the formulation. The osmolarity of the nanosuspension was 303 ± 5 mOsmol/L, which was comparable to that of the tear fluid’s osmolarity (302 mOsmol/L) in normal eyes [37]. The viscosity of LZ–CSNPs was found to be 21.75 cPs (at 35 °C), which was equivalent to the optimum viscosity (which is 20 cPs) that human eyes can tolerate easily without any blurring or difficulty in vision [36].

### 3.4. In Vitro Drug Release and Kinetics of Drug Release

The in vitro release of LZ from CSNPs and LZ–AqS through the dialysis membrane as a release barrier was good in the release medium (STF). The drug release profiles (Figure 3), representing around 80% of the drug was released from the LZ–AqS within 2 h, while almost equal amount (i.e., 79.8%) of the drug was released at 12 h from the LZ–CSNPs and the release of LZ from CSNPs was in a sustained pattern. After assessing the release profiles of LZ from CSNPs, it was seen that it took 24 h to release around 91% of the drug while from its counter formulation (LZ–AqS) around 87% of the drug was released within 4 h. The results suggested that the LZ–loaded CSNPs might be an important means for the sustained ocular delivery of LZ after topical application. The sustained drug release characteristic of the CSNPs could be helpful in maintaining the therapeutic index of the released LZ for a long period with less dosing.

The application of release models to check the drug release kinetics further suggested the sustained release property of the CSNPs [11,16] as shown in Appendix A. Appendix A represents the square root of time vs. the fraction of released drug (Higuchi–Matrix Model). After the application of the kinetic models, it was seen that the release of LZ from CSNPs primarily followed Higuchi’s Square Root pattern (Higuchi–Matrix Model). The regressed line was showing almost linear and crossed very near to the origin of the plot with the highest correlation coefficient value (*R*^2^ = 0.9895) among the applied models. The linearity as evidenced in Higuchi’s square root model further designated the sustained drug release characteristic of CSNPs.

With the help of R2 and the values of slopes of kinetic model equations, the release exponents (*n*-value) were calculated. The values of slope and R2 of the plots of different kinetic equations with the release exponents (*n*-values) are summarized in Table 2. The sustained drug release from the CS–based polymeric matrix occurs by means of three mechanisms, such as (i) liberation of the drug from the matrix because of matrix erosion, (ii) drug diffusion across the matrix, and (iii) a combination of (i) and (ii) mechanisms [52,57,58]. The release pattern of LZ from LZ–loaded CSNPs in this study suggested that the bio–degradation and erosion of CS were the reasons for the sustained release of LZ, which also could control the pattern of release up to 24 h of the release experiment.

The release exponent value (0.0763) calculated for the Higuchi–Matrix Model suggested that the release of LZ from CSNPs followed the “Fickian–Diffusion mechanism. Along with the Higuchi–Matrix Model, the second best–fit model for the release of LZ from CSNPs was the Hixson–Crowell Cube–Root based on the second highest value of correlation efficiency (*R*^2^ = 0.9791) and the mechanism was again the “Fickian–Diffusion”.

### 3.5. Antimicrobial Activity of LZ–CSNPs

The antimicrobial susceptibility results were summarized in Table 3. As compared to LZ–AqS, the LZ–CSNPs presented significantly (*p* < 0.05) improved antimicrobial activities against the employed Gram–positive strains such as *B. subtilis*, *S. aureus*, MRSA (SA–6538) and *S. pneumoniae*, while the blank–CSNPs could not show such activities (Figure 4).

LZ–AqS (µg/mL) demonstrated potent in vitro activity against a set of B. subtilis, MRSA (SA 6538), S. aureus, and S. pneumoniae isolates, the MIC is 5.33 ± 1.15, 8.0 ± 2.0, 5.33 ± 1.0 and 4.0 ± 0.0 µg/mL as shown in Appendix A. However, MIC of LZ–CSNPs (µg/mL) significantly decreased against a set of B. subtilis, MRSA (SA 6538), S. aureus and S. pneumoniae isolates, the MIC 4.0 ± 2.0, 6.67 ± 1.15, 4.67 ± 1.15 and 4.0 ± 0.0 µg/mL respectively as shown in Appendix A.

Some minor but unnoticeable activities were noted with blank CSNPs which might be due to the natural antimicrobial property of chitosan (CS) due to the presence of cationic amine with primary and secondary hydroxyl (–OH) functional groups in its molecular structure, which was also evaluated previously [53].

The level of significance (*p* < 0.05) between the two LZ–containing formulations and that of the blank–CSNPs was further evaluated by One–way analysis of variance (ANOVA) followed by “Tukey’s Multiple Comparison Test (GraphPad Prism V–5)” and data were summarized in Table 3. The increased activity of LZ–CSNPs > LZ–AqS as compared to blank–CSNPs pointed out that the process parameters and formulation steps did not affect the intrinsic antimicrobial property as well as the structural activity relationship of linezolid (LZ). Thus, the encapsulation of LZ into CS–based NPs would not only improve the ocular bioavailability of LZ but also improve and maintain its antimicrobial effectiveness.

### 3.6. Stability Study

The results obtained for the selected physical stability parameters were summarized in Table 4. The results did not demonstrate any noticeable changes in the selected parameters except a slight increase in the particle size at three months and six months at both storage conditions. The increase in the size of the nanoparticles (although not significant (*p* < 0.05)) was attributed to the aggregation nature of CS due to the loss of protection ability of the cryoprotectant. Such particle growth might be associated with the moisture adsorption phenomenon of CSNPs, as there might be traces of moisture present in the dried samples. A similar finding was also noted during the storage of drug–loaded CSNPs in previous reports [13,59].

No significant (*p* < 0.05) changes in the PDI, ZP, EE%, and DL% were found even at raised temperature conditions (40 °C) for six months. Therefore, the LZ–CSNPs were physically stable in terms of particle size, PDI, ZP, EE%, and DL% at the two storage temperatures (25 °C and 40 °C). Thus, we conclude that the developed LZ–CSNPs can be stored for six months at the above conditions without any significant fluctuations in the above–selected parameters.

### 3.7. Eye Irritation Study

The scores for the acute irritation test after topical application of LZ–CSNPs, LZ–AqS, and CSNP have been mentioned (Table 5).

No apparent signs and symptoms of uneasiness were observed in the animals after treatment with the LZ–CSNPs, LZ–AqS, and blank CSNP. Figure 5a,a’,a” represents the images of saline–treated eyes of the respective groups (Green arrows). After 1 h of instillation of LZ–CSNPs (Figure 5b), LZ–AqS (Figure 5b’), and CSNP (Figure 5b”) there was mild redness (Red arrow) with mild abnormal discharge but not any conjunctival inflammation (Red arrows). The redness disappeared at 3 h in the case of LZ–CSNPs and CSNPs treated eyes (Figure 5c,c”, Black arrow) but continued in the case of LZ–AqS treated eyes with slight watery discharge (Figure 5c’, Red arrow). Such symptoms completely disappeared at 6 h (Figure 5d) and the eyes become normal (Green arrow). The redness disappeared but there was a minor watery discharge at 6 h (Figure 5d’) in the case of the LZ–AqS treated eye (Black arrow). Afterward, all abnormal signs and symptoms disappeared at 12 h (Figure 5e,e’,e”) and continued further at 24 h (Figure 5f,f’,f”), and the treated eyes regained their normal condition (Green arrows). Conclusively, the LZ–CSNPs treated eyes did not show any irritation signs at different time intervals, except for some minor symptoms at 1 h (Red arrow). After 6 h onwards, the appearance of normal eye conditions in all LZ–CSNPs, LZ–AqS, and CSNPs treated animals was attributed to the inherent ocular defensive mechanisms. The presence of LZ (broad–spectrum antibiotic) might help the treated eyes to regain their normal conditions. Moreover, the fast recovery of the LZ–CSNPs treated eyes suggested that the excipients (TPP and CS) present in the formulation were biocompatible and non–irritant and the product has good ocular tolerance properties.

The application of LZ–AqS resulted in minor irritation as mentioned above in one out of four treated rabbits with some watery discharge and was designated as one (score–1) and no corneal opacity was noted in the treated eyes, hence, for cornea, iris, and conjunctiva zero (score–0). Considering the classification systems for eye irritation [48,49], the MMTS was calculated (Appendix A). The application of CSNPs resulted in minor irritation as mentioned above in one out of four treated rabbits with some watery discharge and was designated as one (score–1) no corneal opacity was noted in the treated eyes, hence, for cornea, iris, and conjunctiva zero (score–3.5). At 24 h, the MMTS for LZ–CSNPs, LZ–AqS, and CSNPs were 6.75, 14.5, and 8.5 respectively (Table 6. All these values were > 2.6 but < 15, hence, the formulations (LZ–AqS and LZ–CSNPs) including the blank CSNPs were “minimally irritating” and well tolerated by rabbit eyes. The low MMTS values signify the merits of LZ–containing formulations for ophthalmic use.

### 3.8. Transcorneal Permeation of LZ from CSNPs

We could not supply the nutrients to the excised corneal during this experiment, so it was completed for up to 4 h only. From the plots of the amount permeated vs. time (Figure 6), the permeation parameters were calculated and briefed (Table 7). The LZ–loaded CSNPs indicated the linear permeation of LZ more than that of the LZ–AqS. However, the total amounts of drug permeated at 4 h were 56.84 ± 2.33 and 47.71 ± 2.78 µg/cm^2^ from LZ–CSNPs and LZ–AqS, respectively. Although, there was no significant difference (only a 1.2-fold increase) in the total amounts of permeated drug the pattern of permeation profile was entirely different, as a significantly high amount of LZ (31.73 ± 2.61 µg/cm^2^) was crossed the excised cornea from the AqS in 1 h as compared to CSNPs (where it was only 16.03 ± 2.15 µg/cm^2^). Likewise, 42.87 ± 3.58 µg/cm^2^ of LZ was passed from LZ–AqS at 2 h, while it has taken 3 h from CSNPs to permeate approximately equal amount of the drug (43.18 ± 2.53 µg/cm^2^). The increased permeation of LZ from the two formulations was attributed to the occurrence of a unionized form of LZ at the pH conditions of the products (approximately, pH 7). Being a weak base the LZ has a pKa value of 1.8, which indicates that LZ was not ionized in the aqueous environment and the unionized form of the drug could easily permeate across the biological membrane [60].

Overall, a 4.5-fold increased flux and apparent permeability of LZ was achieved by CSNPs as compared to LZ–AqS. From the pattern and shape of permeation profiles, we could say that the LZ–CSNPs provide a sustained release and permeation of the loaded drug that of the conventional aqueous suspension of LZ. Moreover, from the pattern of permeation and the obtained values of parameters, we expect that CSNPs would give a sustained delivery of LZ in the eyes after topical application, which ultimately enhances the ocular availability of the drug by such a non–invasive route of administration.

### 3.9. Ocular Pharmacokinetics

The chromatographic separation of LZ was completed by adopting the reported HPLC method that was successfully used for LZ quantification in the AqH samples [31]. The drug concentrations in AqH were plotted against time (Figure 7) and pharmacokinetic parameters were calculated for both the LZ–containing formulations (Table 8). Initially, the topical instillation of LZ–AqS resulted in faster release and absorption of LZ than that of the LZ–CSNPs and achieved the maximum concentration (C_max_) of 641.98 and 715.96 ng/mL, respectively at the T_max_ of 2 h (LZ–AqS) and 4 h (LZ–CSNPs). Afterward, the concentration of LZ in the AqH was decreasing in a linear way with the mean elimination half–lives of 5.16 h (LZ–AqS) and 8.73 h (LZ–CSNPs). In the case of LZ–AqS the highest drug concentration was found at 2 h which indicated that the LZ was absorbed rapidly up to 2 h only, after that the drug concentration was decreased at further time points than that of the LZ–CSNPs. This phenomenon was supported by the results observed during the transcorneal permeation of LZ from LZ–AqS, where initially it was faster, but a sustained permeation of LZ was observed from LZ–CSNPs during transcorneal permeation and the same observation was noted in ocular bioavailability study also for LZ–CSNPs.

Other than C_max_, significant improvements (*p* < 0.05) were found in the other pharmacokinetic parameters for LZ–CSNPs. Around 1.7–times enhanced t_1/2_ (h) was found for CSNPs as compared to its counter formulation (LZ–AqS). Additionally, 2.7-, 3.1-, 6.2- and 1.9-fold enhanced AUC_0–24h,_ AUC_0–∞,_ AUMC_0–∞,_ and MRT_0–∞_, respectively were found in the case of LZ–CSNPs as compared to LZ–AqS. The clearance rate (CL/F) of LZ was 3.1-times faster than AqS as compared to CSNPs. The rapid clearance of the drug was evidenced by the relatively shorter half–life (t_1/2_; 5.16 h) of LZ and shortened MRT_0–∞_ (6.98 h) of AqS as compared to a prolonged half–life (t_1/2_; 8.73 h) of LZ and MRT_0–∞_ (13.92 h) of LZ–CSNPs. This was also confirmed by the rapid elimination rate of LZ from AqS as a very small concentration of LZ (18.2 ng/mL) was detected at 24 h from AqS. The rapid clearance of LZ from AqS form was the main reason for the low ocular availability of the drug from this formulation.

The relatively improved ocular availability of LZ from CSNPS was due to its prolonged MRT. The prolonged MRT of CSNPs was attributed to its high positive surface charges (due to cationic functional groups (amine) of CS)), which electrostatically interacted with the negatively charged ocular mucin layer. This interaction helped in better penetration of CSNPs across the corneal and conjunctival layer, increased cellular uptake, and enhanced the bioavailability of LZ [16,61,62]. The smaller size of CSNPs could be another important reason for its enhanced transcorneal passage, which was also reported for improved ocular availability of hydrocortisone from the nanosuspension as compared to its micro–sized formulations [35,63]. Finally, the CSNPs was having prospective of improved ocular availability of LZ at small topical doses and reduced dosing frequency of LZ–CSNPs.

## 4. Conclusions and Future Prospective

The ionic–gelation method for the preparation of CSNPs has shown sufficient encapsulation (71%) and loading (11.2%) of LZ into the NPs. of CS and TPP. A sustained release of LZ (90% till 24 h) was found from the NPs in STF through the dialysis membrane. The release of LZ from CSNPs primarily followed Higuchi’s square root model followed by Hixson–Crowell cube–root model release–exponents of 0.076 and 0.015, respectively, which suggested the mechanism of drug release was Fickian–diffusion. The LZ–CSNPs have shown 1.4 to 1.6-times increased antibacterial activity against Gram–positive microbes with the highest inhibition (36.9 mm) against *S. aureus*. The eye irritation test suggested that the LZ–CSNPs were minimally irritating (good ocular tolerance) to rabbit eyes. As compared to LZ–AqS around 4.48-times enhanced transcorneal flux and apparent permeability with 1.2-fold improved drug permeation were noted from LZ–CSNPs. Approximately 2.7-, 3.1-, 6.2- and 1.9-times increased AUC_0–24h_, AUC_0–∞_, AUMC_0–∞_, and MRT_0–∞_, respectively were achieved for LZ from CSNPs as compared to LZ–AqS. Around 1.7-times enhanced half–life of LZ was noted from CSNPs. The increased transcorneal flux and improved ocular bioavailability of LZ from CSNPs indicated that the positively charged CSNPs could be promising nanocarriers for efficient ocular delivery of LZ. The LZ–CSNPs could have the potential for topical application in the treatment of eye infections and related inflammatory conditions due to MRSA and other Gram–positive pathogens. The risk of linezolid resistance due to frequent intravitreal injection during the treatment of endophthalmitis (one of the sight–threatening conditions) can be avoided by the topical application of LZ–CSNPs. Therefore, LZ–loaded CSNPs can be a good alternative to intravitreal injection therapy of linezolid in such eye conditions.

Additional in vivo pharmacodynamics including the uveitis model in rabbits might be required to check the antibacterial activity of LZ in the anterior/posterior eye segment infections and other retinal conditions after topical administration of LZ–loaded CSNPs. Some advanced research works are required to validate the safety and efficacy of CSNPs for clinical applications in human trials.

## Figures and Tables

**Figure 1 nanomaterials-13-00681-f001:**
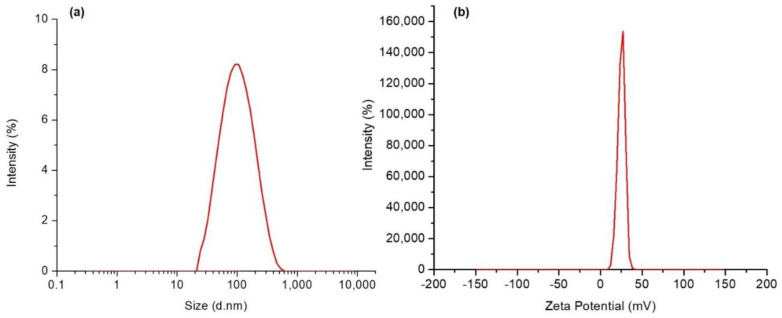
The particle size (**a**) and zeta potential (**b**) distributions of the LZ–CSNPs.

**Figure 2 nanomaterials-13-00681-f002:**
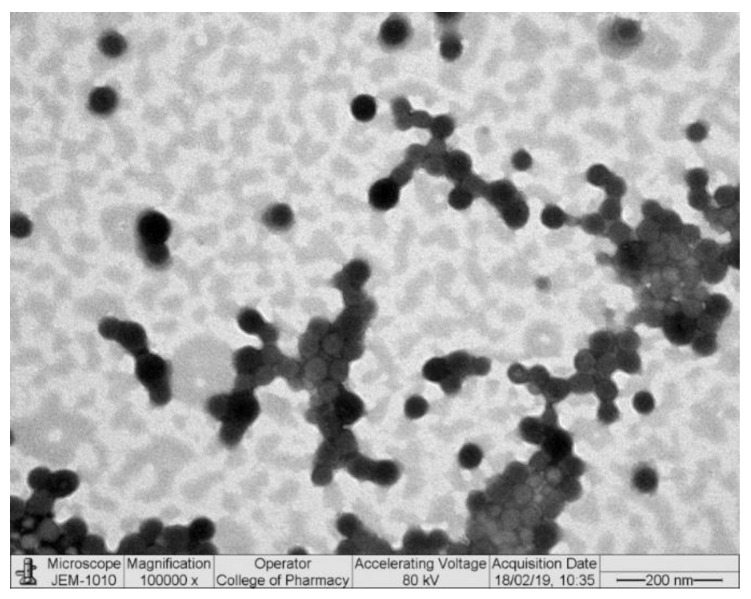
The morphological observation of LZ–CSNPs by Transmission Electron Microscopy. The imaging of the NPs was carried out at 100,000–times magnification and the scale bar size was 200 nm.

**Figure 3 nanomaterials-13-00681-f003:**
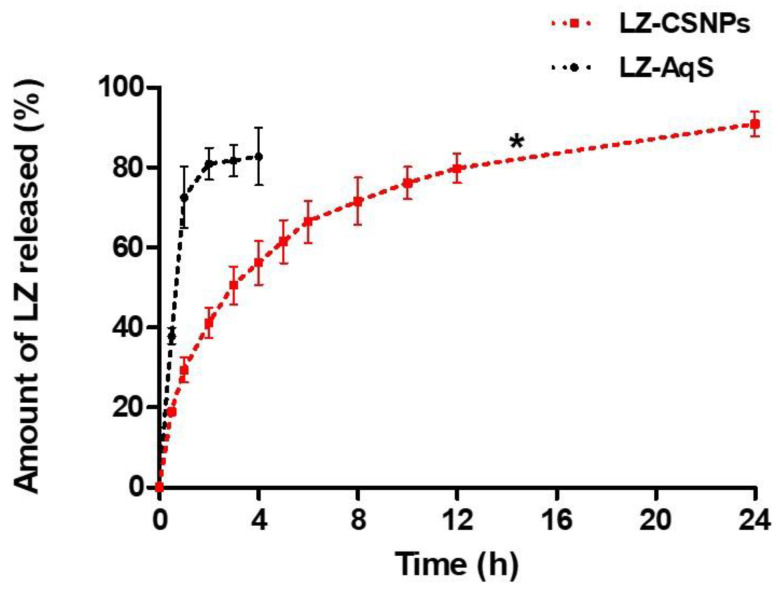
In vitro release of LZ from CSNPs and LZ–AqS in simulated tear fluid (STF) at pH 7.4 through the dialysis membrane. Results were represented as the mean of three measurements with standard deviation (Mean ± SD), The “*” representing *p* < 0.05 was the level of significance between the two drug products.

**Figure 4 nanomaterials-13-00681-f004:**
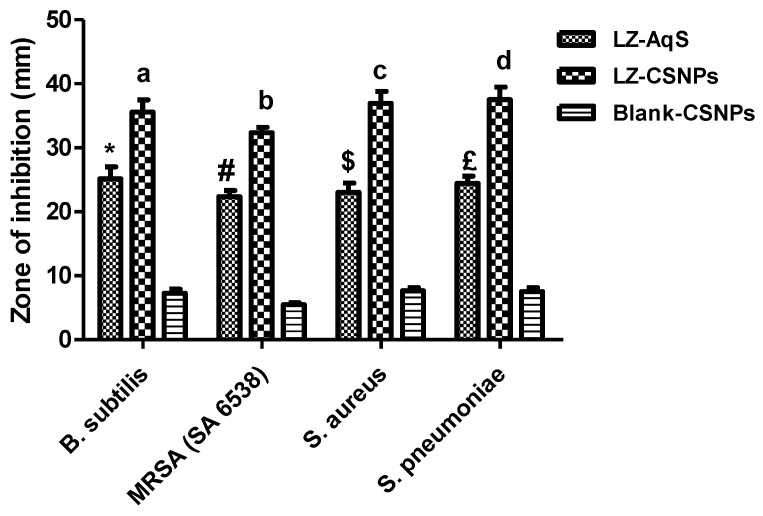
Antimicrobial activity of LZ–containing products as compared to the blank CSNPs against *Bacillus subtilis*, MRSA (SA–6538), *Staphylococcus aureus,* and *Streptococcus pneumoniae.* Results were represented as mean of three measurements with standard deviation (Mean ± SD). “a” *p* < 0.05, LZ–CSNPs vs. other two products (for *B. subtilis*); “b” *p* < 0.05, LZ–CSNPs vs. other two products (for MRSA; SA–6538); “c” *p* < 0.05, LZ–CSNPs vs. other two products (for *S. aureus*); “d” *p* < 0.05, LZ–CSNPs vs. other two products (for *Streptococcus pneumoniae*). Additionally, “*” *p* < 0.05, LZ–CSNPs vs. Blank–CSNPs (for *B. subtilis*); “#” *p* < 0.05, LZ–CSNPs vs. Blank–CSNPs (for MRSA; SA–6538); “$” *p* < 0.05, LZ–CSNPs vs. Blank–CSNPs (for *S. aureus*); “£” *p* < 0.05, LZ–CSNPs vs. Blank–CSNPs (for *Streptococcus pneumoniae*).

**Figure 5 nanomaterials-13-00681-f005:**
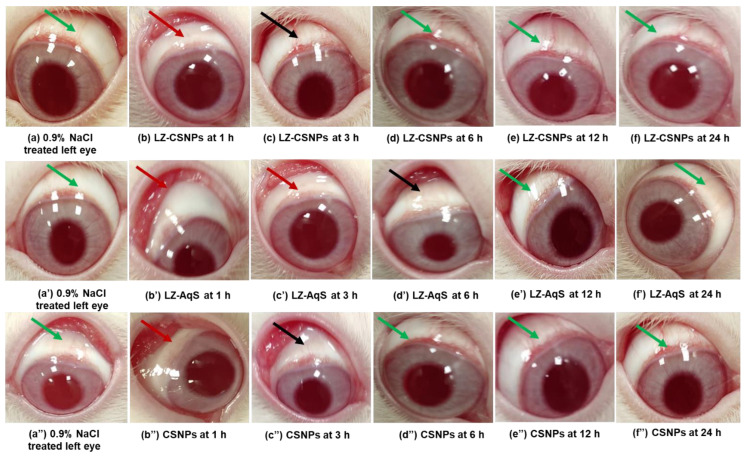
Eye images captured during irritation study. Representative images of normal saline treated eyes (**a**), (**a’**) and (**a”**) (green arrows). Post topical application of LZ–CSNPs at 1 h (**b**); (red arrows) at 3 h (black arrows) (**c**); at 6 h (**d**); at 12 h (**e**) and at 24 h (**f**), (green arrow) whereas the post topical application of LZ–AqS at 1 h (**b’**) (red arrows); at 3 h (**c’**) red arrows); at 6 h (**d’**); at 12 h (**e’**) and at 24 h (**f’**) (green arrow) and post topical application of CSNPs at 1 h (**b”**) (red arrows); at 3 h (**c”**) (black arrows); at 6 h (**d”**); at 12 h (**e”**) and at 24 h (**f”**) (green arrow).

**Figure 6 nanomaterials-13-00681-f006:**
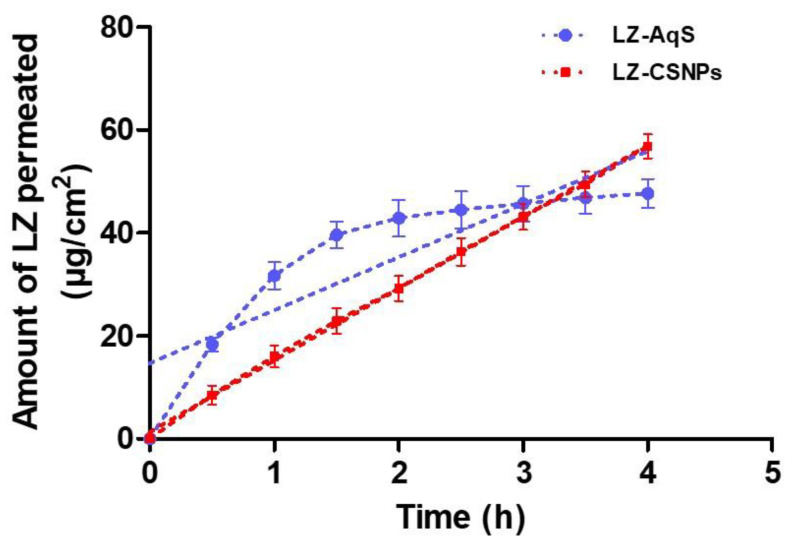
Transcorneal permeation Transcorneal permeation of LZ from CSNPs and LZ–AqS. Results were represented as the mean of three measurements with standard deviation (Mean ± SD, *n* = 3).

**Figure 7 nanomaterials-13-00681-f007:**
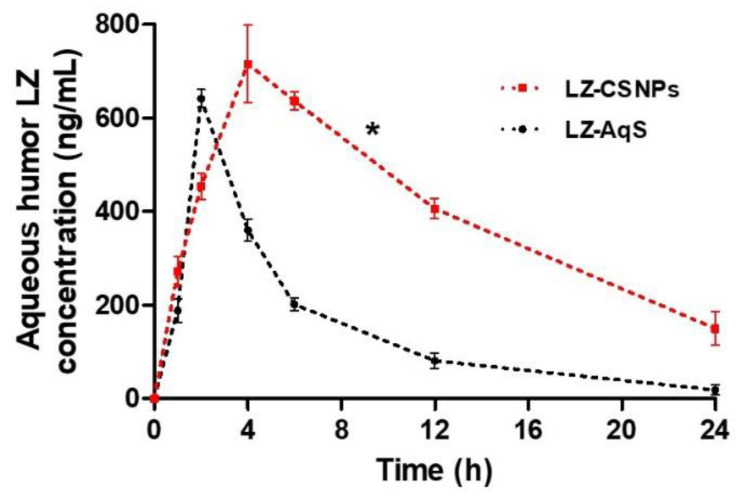
The drug concentration–time profile of aqueous humor (AqH) samples collected from rabbit eyes post topical application of LZ–AqS and LZ–CSNPs. Results were represented as the mean of three readings (three rabbits per group) with standard error of mean (Mean ± SEM). The “*” representing *p* < 0.05 was the level of significance between the two drug products.

**Table 1 nanomaterials-13-00681-t001:** Physicochemical characteristics of nanosuspension of LZ–containing formulations as compared to STF. Results were presented as the mean of three measurements with standard deviation (Mean ± SD, *n* = 3).

Formulation	Clarity at 25 °C	pH	Osmolarity (mOsmol/L)	Viscosity (cPs)	
				At 25 °C	At 35 °C
LZ–CSNPs	Transparent	7.3 ± 0.4	303 ± 5	22.45 ± 3.24	21.75 ± 2.97
LZ–AqS	Transparent	7.1 ± 0.5	302 ± 3	2.56 ± 0.94	2.25 ± 1.06
STF *	Transparent	7.4 ± 0.2	301 ± 4	2.05 ± 0.05	2.09 ± 0.04

* STF stands for simulated tear fluid and cPs stands for Centipoises (unit of viscosity).

**Table 2 nanomaterials-13-00681-t002:** Different release kinetics model equations with release exponents (*n*-values).

Applied Kinetic Models	*R*^2^ Values	Slope	*n*-Values
Zero order (Fraction drug released vs. Time)	0.7912	0.0207	0.0089
First order (Log% Drug remaining vs. Time)	0.9517	0.0379	0.0164
Korsmeyer–Peppas (Log Fraction drug released vs. log Time)	0.9615	0.3639	0.1580
Higuchi Matrix (Fraction drug released vs. Square root of Time)	0.9895	0.1758	0.0763
Hixon–Crowell (M_o_^1/3^–M_t_^1/3^ vs. Time)	0.9791	0.0349	0.0152

**Table 3 nanomaterials-13-00681-t003:** The zone of inhibitions achieved through the agar diffusion test by LZ–CSNPs as compared to LZ–AqS. Results were represented as the mean of three measurements with standard deviation (Mean ± SD).

Microorganisms	Zone of Inhibition by (mm), Mean ± SD, *n* = 3			
	LZ–CSNPs	LZ–AqS	Blank–CSNPs	Fold Increase
*Bacillus subtilis*	35.57 ± 2.69	25.17 ± 2.62	7.26 ± 0.87	1.41–times
*MRSA (SA 6538)*	32.36 ± 1.14	22.36 ± 1.32	6.47 ± 0.13	1.60–times
*Staphylococcus aureus*	36.93 ± 2.65	23.03 ± 2.04	7.65 ± 0.67	1.54–times
*Streptococcus pneumoniae*	37.52 ± 2.73	24.45 ± 1.61	7.53 ± 0.86	1.45–times
**Statistical analysis by ANOVA**				
**Tukey’s Multiple Comparison Test**	**Mean Difference**	*** q = (D/SED)**	***p* < 0.05**	**95% CI * Difference**
LZ–AqS vs. LZ–CSNPs	−11.85	14.51	Yes	−15.08 to −8.62
LZ–AqS vs. Blank–CSNPs	16.77	20.53	Yes	13.54 to 19.99
LZ–CSNPs vs. Blank–CSNPs	28.62	35.03	Yes	25.39 to 31.84

* Where q = Square Root, SED = Standard error of the difference, D = Difference between two means, and CI = Confidence interval.

**Table 4 nanomaterials-13-00681-t004:** Stability study results for LZ–CSNPs stored at 25 °C and 40 °C for six months. Data were presented as the mean of three measurements with standard deviations (Mean ± SD, *n* = 3).

Parameters (Mean ± SD)	Time Points				
	Initially (0 h)	At 7th Day	At 1st Month	At 3rd Month	At 6th Month
**At 25 °C storage temperature**					
**Particle size (nm)**	213.73 ± 10.91	215.13 ± 6.97	216.3 ± 8.67	217.93 ± 8.61	219.13 ± 9.42
**PDI**	0.387 ± 0.134	0.392 ± 0.124	0.395 ± 0.125	0.401 ± 0.118	0.418 ± 0.105
**Zeta Potential (mV)**	23.13 ± 2.99	22.43 ± 3.06	22.13 ± 3.02	21.73 ± 3.03	21.47 ± 3.07
**EE (%) ***	71.05 ± 5.22	70.59 ± 5.24	70.15 ± 5.16	69.52 ± 4.77	69.09 ± 4.56
**DL (%) ***	11.19 ± 1.82	11.11 ± 1.29	11.05 ± 1.08	10.94 ± 0.97	10.88 ± 1.22
**At 40 °C storage temperature**					
**Particle size (nm)**	213.73 ± 10.91	215.40 ± 9.14	217.21 ± 8.91	219.53 ± 9.30	223.37 ± 12.32
**PDI**	0.387 ± 0.134	0.393 ± 0.124	0.397 ± 0.125	0.411 ± 0.122	0.429 ± 0.107
**Zeta Potential (mV)**	23.13 ± 2.99	22.27 ± 2.96	21.9 ± 3.09	21.0 ± 3.17	20.6 ± 3.14
**EE (%) ***	71.05 ± 5.23	70.21 ± 4.39	69.53 ± 4.45	69.38 ± 4.76	68.25 ± 5.64
**DL (%) ***	11.18 ± 1.82	11.06 ± 1.69	10.95 ± 1.71	10.93 ± 1.09	10.74 ± 1.08

* Where, EE (%) = Percentage of encapsulation efficiency and DL (%) = Percent drug loading.

**Table 5 nanomaterials-13-00681-t005:** Weighted scores for irritation test of LZ–CSNPs, LZ–AqS and CSNPs in rabbit eyes.

Lesions in the Treated Eyes	Individual Scores of Eye Irritation											
	For LZ–CSNPs				For LZ–AqS				CSNPs			
	Animal Numbers				Animal Numbers				Animal Numbers			
	1st	2nd	3rd	4th	1st	2nd	3rd	4th	1st	2nd	3rd	4th
In Cornea												
(a) Opacity (Degree of density)	1	0	0	0	0	1	0	1	0	0	0	0
(b) Area of cornea	4	4	4	4	4	4	4	4	4	4	4	4
Total scores = (a × b × 5) =	20	0	0	0	0	20	0	20	0	0	0	0
In Iris												
(a) Lesion values	1	0	0	0	0	1	0	1	0	1	1	0
Total scores = (a × 5) =	5	0	0	0	0	5	0	5	0	5	5	0
In Conjunctiva												
(a) Redness	1	0	0	0	0	1	0	1	0	1	1	0
(b) Chemosis	0	0	0	0	0	0	0	0	0	0	0	0
(c) Mucoidal discharge	0	0	0	0	0	1	0	1	0	0	0	0
Total scores = (a + b + c) × 2 =	2	0	0	0	0	4	0	4	0	2	2	0

**Table 6 nanomaterials-13-00681-t006:** The calculation of the Maximum Mean Total Score (MMTS) based to the obtained scores as mentioned above.

For LZ–CSNPs						
	Animal Numbers				SUM	Average (SUM/4)
	1st	2nd	3rd	4th		
Cornea	20	0	0	0	20	5.00
Iris	5	0	0	0	5	1.25
Conjunctiva	2	0	0	0	2	0.50
SUM total =	27	0	0	0	27	6.75
**For LZ** **–AqS**						
	**Animal Numbers**				**SUM**	**Average (SUM/4)**
	**1st**	**2nd**	**3rd**	**4th**		
Cornea	0	20	0	20	40	10.00
Iris	0	5	0	5	10	2.50
Conjunctiva	0	4	0	4	8	2.00
SUM total =	0	29	0	29	58	14.50
**For Blank CSNP**						
	**Animal Numbers**				**SUM**	**Average (SUM/4)**
	**1st**	**2nd**	**3rd**	**4th**		
Cornea	0	20	0	0	20	5.00
Iris	0	5	5	0	10	2.50
Conjunctiva	0	2	0	2	4	1.00
SUM total =	0	27	5	2	34	8.50

**Table 7 nanomaterials-13-00681-t007:** Transcorneal permeation parameters from LZ–CSNPs and its counter formulation (LZ–AqS). The results are the mean of three measurements with standard deviations (Mean ± SD, *n* = 3).

Permeation Parameters	Formulations		Enhancement
	LZ–AqS	LZ–CSNPs	
Cumulative amount of LZ permeated (µg/cm^2^)	47.71 ± 2.78	56.84 ± 2.33	1.20-fold
Steady–state flux, *J* (µg/cm^2^/h)	6.068 ± 1.46	27.191 ± 1.69	4.48-fold
Apparent permeability, *P_app_* (cm/h)	(0.606 ± 0.15) × 10^−2^	(2.72 ± 0.01) × 10^−2^	4.48-fold

**Table 8 nanomaterials-13-00681-t008:** Ocular pharmacokinetics of LZ after topical application of LZ–containing formulations. Data were represented as the mean of three measurements with standard error of mean (Mean ± SEM, *n* = 3).

Pharmacokinetic Parameters	LZ–Containing Formulations (Mean ± SEM, *n* = 3)		Enhanced Ratios
	LZ–AqS	LZ–CSNPs	
t_1/2_ (h)	5.16 ± 0.66	8.73 ± 1.94 *	1.7
T_max_ (h)	2.00 ± 0.00	4.00 ± 0.00 *	2.0
C_max_ (ng/mL)	641.98 ± 10.94	715.96 ± 47.38	1.2
AUC_0–24 h_ (ng/mL·h)	3517.61 ± 198.79	9489.91 ± 48.51 *	2.7
AUC_0–inf_ (ng/mL·h)	3664.81 ± 267.24	11433.67 ± 471.41 *	3.1
AUMC_0–inf_ (ng/mL·h^2^)	25965.93 ± 4582.76	160532.91 ± 22002.64 *	6.2
MRT_0–inf_ (h)	6.98 ± 0.71	13.92 ± 1.35 *	1.9
Cl/F (mL/h)	13.78 ± 1.95 *	4.39 ± 1.18	3.1

Where “*” represents the level of significance (*p* < 0.05).

## Data Availability

The data presented in this study are available on request from the corresponding author.

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
