# Peer review of "Topical Application of Linezolid–Loaded Chitosan Nanoparticles for the Treatment of Eye Infections"

_nanomaterials, 2023, doi:10.3390/nano13040681_

Round 1

Reviewer 1 Report

Alkholief and colleagues designed linezolid-loaded chitosan nanoparticles and investigated their potential applications in eye treatment. In this study, authors showed promising data to display the superiority of this nanoparticle versus aqueous solution. Some comments are still needed to improve this manuscript.

The detailed comments are listed:

1 the nanoparticle in TEM looks small and is contradict to the data from DLS. Authors have to claim that discrepancy.

2 the font in this manuscript is not consistent among tables and text, which has to be adjusted and optimized.

3 antibacterial MIC of this nanoparticles should be provided and compared.

4 irritation of the blank nanoparticle needs to be tested.

Author Response

Comments and Suggestions for Authors by Reviewer #1

Alkholief and colleagues designed linezolid-loaded chitosan nanoparticles and investigated their potential applications in eye treatment. In this study, authors showed promising data to display the superiority of this nanoparticle versus aqueous solution. Some comments are still needed to improve this manuscript.

The detailed comments are listed:

1 the nanoparticle in TEM looks small and is contradict to the data from DLS. Authors have to claim that discrepancy.

Response:

Dynamic light scattering (DLS) works on the fundamentals, of Brownian motion (a random thermal motion associated with fine particles and molecules) which is temperature dependent, therefore the DLS instruments are equipped with precision temperature control. The DLS measures the diameter of hydrated (solvated) particles by analyzing the modulation of scattered light intensity as a function of time.[1] The particle radius is inversely proportional to the diffusion coefficient and calculated by the inbuilt software using the Stokes-Einstein equation [2]

Where kB is the Boltzmann constant, T is the Temperature, ƞ is the viscosity and R is the radius.

The smaller particles will diffuse faster than the larger particles and their diffusion coefficient will be higher. The TEM measures the diameter of bare particles. Thus the particle size from the DLS is comparatively larger than the TEM. Further, the hydrodynamic diameter is dependent on the surface charge density, the higher the charge density higher will be the deviation [3]. The ionic strength of the medium is another factor influencing the hydrodynamic diameter. A low-conductivity medium (Distilled water) has the tendency to produce an extended double layer which will increase the hydrodynamic particle size while the comparatively higher-conductivity medium will suppress the electrical double layer resulting in reduced hydrodynamic diameter [4]. Another difference associated with DLS is that it works on a comparatively larger population of particles with cumulant analysis which provides the intensity-weighted mean diameter (Z-Average Diameter) [5] while the TEM deals with relatively very small sample sizes and appropriately diluted samples provide a good image of properly dispersed and non-overlapping particles.

[1] T.G. Souza, V.S. Ciminelli, N.D.S. Mohallem, A comparison of TEM and DLS methods to characterize size distribution of ceramic nanoparticles,  Journal of physics: conference series, IOP Publishing, 2016, pp. 012039.

[2] C.M. Hoo, N. Starostin, P. West, M.L. Mecartney, A comparison of atomic force microscopy (AFM) and dynamic light scattering (DLS) methods to characterize nanoparticle size distributions, Journal of Nanoparticle Research, 10 (2008) 89-96.

[3] M. Rezaei, A.R. Azimian, A.R. Pishevar, Surface charge-dependent hydrodynamic properties of an electroosmotic slip flow, Physical Chemistry Chemical Physics, 20 (2018) 30365-30375.

[4] J. Radermacher, The effect of ionic strength on hydrodynamic radius for different microparticle surfaces, (2012).

[5] N. Farkas, J.A. Kramar, Dynamic light scattering distributions by any means, Journal of Nanoparticle Research, 23 (2021) 120.

2 the font in this manuscript is not consistent among tables and text, which has to be adjusted and optimized.

Response: As per the suggestion, the font size and style in the texts and Tables have been adjusted and optimized.

3 antibacterial MIC of this nanoparticles should be provided and compared.

Response: The MIC values of LZ-AqS and LZ-CSNPs against different microorganism were incorporated in the manuscript under section 3.5 Antimicrobial activity of LZ-CSNPs and highlighted with yellow colour. A table (Table S2) for the same included in the supplementary materials.

4 irritation of the blank nanoparticle needs to be tested.

Response: As per the suggestion the ocular irritation test of blank nanoparticles (CSNPs) was performed and the weighted scores for the irritation test of CSNPs were incorporated in the Table 5. The representative figures of the irritation testing of CSNPs were included in Figure 5 and designated as a”, b”, c”, d”, e” and f” for 0.9% NaCl treated eye, CSNPs treated eyes at 1 h, 3 h, 6 h, 12 h and 24 h respectively. The blank CSNPs ocular irritation study related observations were also added in the manuscript in the section 3.7 Eye irritation study and highlighted with yellow colour.

Reviewer 2 Report

The manuscript « Topical Application of Linezolid-loaded Chitosan Nanoparti-2 cles for the Treatment of Eye Infections» submitted by Musaed Alkholief et al. is devoted to  Linezolid loaded chitosan-nanoparticles (LZ-CSNPs) preparation and investigation of their biophysical properties. The ionic-gelation method was used for the preparation of CSNPs. Authors demonstrated release of LZ from the NPs in STF through the dialysis membrane. The LZ-CSNPs has shown 1.4 to 1.6-times increased antibacterial activity against Gram-positive microbes. The increased transcorneal flux and improved ocular bioavailability of LZ from CSNPs indicated that the positively charged  CSNPs could be the promising nanocarriers for an efficient ocular delivery of LZ.

Abstract reflects the main points of works. The introduction describes the Linezolid (LZ) as the first licensed drug that can be a choice of therapeutic agent to treat of the eye infections. Authors postulate the necessity of design of novel formulation of LZ on the base of Chitosan nanoparticles (CSNP).

In the section material and methods authors carefully described all used methods and approaches. In conclusions authors summarized their results and future prospectives.

I recommend accepting this manuscript for publication minor revision.

I have some questions and remarks:

1.     Line 107. Please change the register for the charges and atom numbers in the formulas (–NH3+) and (–P3O105-).

2.     Lines119-127. Please check the necessity of " in the text.

3.     Please change the celsius degree symbol for °C (Lines 135, 149, 151, 152, 153, 155).

4.     Line 182. Start the sentence with Upper letter.

5.     During the text, several different types of the text format were used. Please used the formatting of the text corresponding to the rules for this type of publication.

6.     In the text, authors refer to supplementary Figure 439 S1, but I didn't found any supplementary materials. Please add necessary files.

Author Response

Comments and Suggestions for Authors by Reviewer #2

The manuscript « Topical Application of Linezolid-loaded Chitosan Nanoparti-2 cles for the Treatment of Eye Infections» submitted by Musaed Alkholief et al. is devoted to Linezolid loaded chitosan-nanoparticles (LZ-CSNPs) preparation and investigation of their biophysical properties. The ionic-gelation method was used for the preparation of CSNPs. Authors demonstrated release of LZ from the NPs in STF through the dialysis membrane. The LZ-CSNPs has shown 1.4 to 1.6-times increased antibacterial activity against Gram-positive microbes. The increased transcorneal flux and improved ocular bioavailability of LZ from CSNPs indicated that the positively charged CSNPs could be the promising nanocarriers for an efficient ocular delivery of LZ.

Abstract reflects the main points of works. The introduction describes the Linezolid (LZ) as the first licensed drug that can be a choice of therapeutic agent to treat of the eye infections. Authors postulate the necessity of design of novel formulation of LZ on the base of Chitosan nanoparticles (CSNP).

In the section material and methods authors carefully described all used methods and approaches. In conclusions authors summarized their results and future prospectives.

I recommend accepting this manuscript for publication minor revision.

Response: The authors are thankful to the reviewer to read our manuscript very minutely and appreciations. Also, thankful to recommend this manuscript for publication after minor revision.

I have some questions and remarks:

  1. Line 107. Please change the register for the charges and atom numbers in the formulas (–NH3+) and (–P3O105-).

Response: As per the suggestion, these were corrected and incorporated in the revised manuscript.

  1. Lines119-127. Please check the necessity of " in the text.

Response: Actually these are very common sentences used for the sources of chemicals and materials. To avoid any similarity with the published papers, the symbol “” were used.

  1. Please change the celsius degree symbol for °C (Lines 135, 149, 151, 152, 153, 155).

Response: The symbol of degree Celsius degree symbol for °C

  1. Line 182. Start the sentence with Upper letter.

Response: Corrected

  1. During the text, several different types of the text format were used. Please used the formatting of the text corresponding to the rules for this type of publication.

Response: Thanks for the suggestion. The text formatting was done as per the journal requirements.

  1. In the text, authors refer to supplementary Figure 439 S1, but I didn't found any supplementary materials. Please add necessary files.

Response: The Supplementary Material was updated and will be supplied during the submission of the revised manuscript. 

Reviewer 3 Report

This manuscript was using chitosan-nanoparticles (CSNPs) loaded Linezolid to enhance a better-prolonged delivery of LZ in the eye treatment. The author did massive work to prove their results. The overall quality of the article is good, and the data is comprehensive. But I have a few questions as below:

1: Why would you choose this ratio (The LZ-loaded CSNPs at TPP/CS weight ratio of 78 mg:162 mg with 21.7 mg of the drug) to do the whole experiment? Did you try another ratio? If so, please show it in the supporting document.

2: About the DLS data, please add the LZ-AqS solution data. Was it chitosan that causes LZ drug nanonization?

3: Please add a significant difference in Figure 4.

4: In figure 6, All the blue points are higher than the red (except the last two points). The vertical axis was the amount of permeation of LZ, I think the LZ-AqS was higher than the CSNPs groups and relative to hours, will also be higher than CSNPs groups (Steady-state flux, J (µ g/cm2/h)). Would you please explain why Table 7 shows the contrary conclusions?

Author Response

Comments and Suggestions for Authors

This manuscript was using chitosan-nanoparticles (CSNPs) loaded Linezolid to enhance a better-prolonged delivery of LZ in the eye treatment. The author did massive work to prove their results. The overall quality of the article is good, and the data is comprehensive. But I have a few questions as below:

 1: Why would you choose this ratio (The LZ-loaded CSNPs at TPP/CS weight ratio of 78 mg:162 mg with 21.7 mg of the drug) to do the whole experiment? Did you try another ratio? If so, please show it in the supporting document.

Response:

Thanks for the reviewer’s comment regarding formulation development. In our previous publication [52] we used design expert software to optimize and select the final weight ratios of the TPP and CS to get the optimal CSNPs. So, we adopted our previous optimized weight ratio of the excipients to prepare the LZ-loaded CSNPs. The LZ-loaded CSNPs at TPP/CS weight ratio of 78 mg:162 mg with 21.7 mg of the drug resulted best primary characterization parameters as compared to other formulations with varying weight ratio of the excipients. The constraints including the minimum particle size with maximum encapsulation efficiency (%EE), drug loading (%DL) and zeta-potential (ZP) were applied for optimization of the LZ-CSNPs. Based on the obtained parameters (Supplementary materials), the LZ-CSNPs (with 78 mg:162 mg with 21.7 mg of LZ) was found to be the best one among the three formulations. Thus, it was selected for further studies. Results and data were represented in supplementary materials as Table S4.

2: About the DLS data, please add the LZ-AqS solution data. Was it chitosan that causes LZ drug nanonization?

Response: Thanks for the reviewer suggestion, the DLS data associated with LZ-AqS was now added in the revised manuscript. The ionic gelation of chitosan with tripolyphosphate as cross linker generates nano-sized particles and the drug molecules were encapsulated into the core of nanoparticles and some molecules gets adsorbed on the surfaces of CSNPs also. 

3: Please add a significant difference in Figure 4.

Response:

As compared to LZ-AqS, the LZ-CSNPs presented significantly (p < 0.05) improved antimicrobial activities against the employed Gram positive strains such as B. subtilis, S. aureus, MRSA (SA-6538) and S. pneumoniae, while the blank-CSNPs could not show such activities (Figure 4). As represented in the caption of Figure 4 “). “a” p < 0.05, LZ-CSNPs versus other two products (for B. subtilis); “b” p < 0.05, LZ-CSNPs versus other two products (for MRSA; SA-6538); “c” p < 0.05, LZ-CSNPs versus other two products (for S. aureus); “d” p < 0.05, LZ-CSNPs versus other two products (for Streptococcus pneumoniae). Also, “*” p < 0.05, LZ-CSNPs versus Blank-CSNPs (for B. subtilis); “#” p < 0.05, LZ-CSNPs versus Blank-CSNPs (for MRSA; SA-6538); “$” p < 0.05, LZ-CSNPs versus Blank-CSNPs (for S. aureus); “d” p < 0.05, LZ-CSNPs versus Blank-CSNPs (for Streptococcus pneumoniae).

4: In figure 6, All the blue points are higher than the red (except the last two points). The vertical axis was the amount of permeation of LZ, I think the LZ-AqS was higher than the CSNPs groups and relative to hours, will also be higher than CSNPs groups (Steady-state flux, J (µ g/cm2/h)). Would you please explain why Table 7 shows the contrary conclusions?

Response: The authors are thankful to the reviewer to notice Figure 6, where all the blue points are higher than the red up to 3 h (except the last two points). The vertical axis was the amount of drug permeated. Initially the drug was permeated rapidly from the AqS form and after 3 h it was almost constant, contrary to this the permeation of drug was increases even after 3 h from the NPs, this indicated the sustained release of drug from CSNPs. To calculate the steady-state flux and apparent permeability, the slope of the linear ascent of the plots were considered. A linear permeation profile was observed from the NPs throughout the experiment till 4 h, while the linearity was observed from 1.5 to 4 h only from the AqS. This was the reason that the flux and apparent permeability were higher for CSNPs as compared to LZ-AqS. Moreover, the cumulative amount of LZ permeated (µg/cm2) was calculated at the end time (4 h) as summarized in Table 7, so the values for CSNPs were higher as compared to the AqS form.

Round 2

Reviewer 1 Report

I am fine with this revision. No more revision is needed.